# Chemical Composition, In Vitro Antioxidant Activities, and Inhibitory Effects of the Acetylcholinesterase of *Liparis nervosa* (Thunb.) Lindl. Essential Oil

**DOI:** 10.3390/biom13071089

**Published:** 2023-07-07

**Authors:** Jiayi Zhao, Ziyue Xu, Peizhong Gao, Xu Liu

**Affiliations:** 1SDU-ANU Joint Science College, Shandong University, Weihai 264209, China; jiayi_zhao123@outlook.com (J.Z.); 202000700291@mail.sdu.edu.cn (Z.X.); 202000700241@mail.sdu.edu.cn (P.G.); 2Marine College, Shandong University, Weihai 264209, China

**Keywords:** *Liparis nervosa*, essential oil, chemical composition, GC-MS, GC-FID, antioxidant, anti-acetylcholinesterase

## Abstract

The present study aimed to investigate the essential oil composition of *Liparis nervosa* (Thunb.) Lindl., grown in China, and to determine its antioxidant and inhibitory effects on acetylcholinesterase. The essential oil was obtained by hydrodistillation, and the chemical compounds were analyzed by GC-MS and GC-FID. We used 2,2′-azino-bis(3-ethylbenzothiazoline-6-sulfonic acid) (ABTS), 2,2-diphenyl-1-picrylhydrazyl (DPPH), and ferric reducing assay power (FRAP) to evaluate the antioxidant activity. The anti-acetylcholinesterase activity of the essential oil was also examined. Sixty-seven compounds were identified, representing 98.50 % of the total essential oil, which was shown to be rich in methyl (9*E*,11*E*)-octadeca-9,11-dienoate (31.69%), n-hexadecanoic acid (15.08%), isopropyl palmitate (12.44%), propyl tetradecanoate (7.20%), tetradecanoic acid (4.01%), 17-octadecynoic acid (3.71%), and pentacosane (2.24%). Its antioxidant ability was analyzed via ABTS (IC_50_ = 721.95 ± 9.93 μg/mL), DPPH scavenging capacity (IC_50_ > 10,000 μg/mL), and the FRAP method (Trolox equivalent antioxidant concentration 39.64 ± 3.38 μM/g). Acetylcholinesterase inhibition effects were evaluated and had an IC_50_ value of 51.96 ± 14.26 μg/mL. The results show that this essential oil has interesting biological potential, encouraging further investigations, especially regarding the mechanisms of action of its antioxidant and anti-acetylcholinesterase activity. This is the first time that the chemical composition, antioxidant activity, and acetylcholinesterase inhibition effects of essential oil from *L. nervosa* have been studied.

## 1. Introduction

The genus *Liparis* of Orchidaceae is extensive. *Liparis* plants, of which there are approximately 320 species around the world, are generally perennial herbs, with most of these species scattering from the tropics and subtropics to temperate and alpine regions [1]. *Liparis nervosa* was selected since it is a widely cultivated commercial medicinal plant. In China, since ancient times, it has been commonly used for medicinal purposes, such as clearing heat fire, cooling the blood, removing toxicity, curing circulatory collapse, and relieving swelling and convulsion [2]. Most of the diseases mentioned above are not caused by a single mechanism but are instead the result of several biological processes. Therefore, traditional herbal medicine is gaining popularity as an approach to treatment.

There is a wealth of scientific evidence supporting the therapeutic effects of plant-derived molecules, and the exploration of natural product repositories has led to the development of some major conventional medicines, such as the anticancer drug Taxol [3]. Essential oils (EOs) are mixtures of volatile organic compounds containing monoterpenes, sesquiterpenes, alcohols, aldehydes, ketones, acids, phenols, ethers, and esters [4]. In addition, previous research has revealed that aromatherapy with various essential oils can alleviate mental health problems, such as sleep disturbances and anxiety, by affecting the inner nervous and circulatory systems [5]. However, there may be some volatile chemical emissions from essential oils, such as acetaldehyde, limonene, and methanol, which could have adverse health effects, such as allergies. Therefore, these oils should be utilized carefully in therapy and other applications [6,7]. Due to their chemical complexity and diversity, EOs have experienced renewed interest in many fields of study; they have interesting physicochemical characteristics and high added value with respect to the environment [8].

Alzheimer’s disease (AD) is a progressive, degenerative, neurological disorder resulting in impaired memory and behavior. Through inflammation, immune response, and mitochondrial damage, imbalances in the in vivo generation of reactive oxygen species (ROS), reactive nitrogen species (RNS), and antioxidative elimination can lead to oxidative stress, which may promote the accumulation of protein-fragment beta-amyloid (Aβ) plaques in the synaptic space. This accumulation of insoluble proteins is recognized as an indicator of neurotoxicity, resulting in neuronal death and, consequently, Alzheimer’s disease [9,10]. Moreover, previous reviews have shown improved cognitive behavior in animals suffering from AD and treated with antioxidants such as curcumin [11]. The plant-derived free-radical scavenger, ferulic acid, has also been demonstrated as a potential therapeutic agent for AD [12]. In addition, considering that acetylcholine is rapidly hydrolyzed and inactivated by cholinesterase, its inhibition couples with the elevation of the neurotransmitter acetylcholine in the synaptic cleft. Hence, cholinesterase is considered a potential therapeutic target with the interference of inhibitory molecules. Accordingly, antioxidant agents and cholinesterase inhibitors are crucial for AD therapeutics [13,14]. Previous research on essential oils’ biological activities also cast new light on their potential therapeutic prospects [15].

Although various studies focus on chemical-compound isolation—including lectin, nervonic acid derivatives, phenylpropanoids, pyrrolizidine alkaloids derivatives, and pyrrolizidine alkaloids, which have anti-fungal, anti-tumor, antioxidant, and α-glucosidase inhibitory properties [16,17,18,19,20], however the chemical composition, antioxidant activities, and acetylcholinesterase inhibitory effects of the EOs of this species remain unexplored. Thus, the present study was carried out to assess the phytochemical composition and biological activities of EO from *L. nervosa*.

## 2. Materials and Methods

### 2.1. Plant Material

Dry leaves and stems of *L. nervosa* were collected from Haikou Town, Chengjiang City, Kunming, Yunnan Province, China (24.52 N, 102.98 E) in November 2021. This plant was authenticated by Prof. Hong Zhao, Marine College, Shandong University. A voucher specimen was deposited at Marine College with the following registration number: EO2138. The plant material was placed in the shade and maintained under refrigeration (4 °C) until EO extraction.

### 2.2. EO Isolation

The dried plant (750 g) of *L. nervosa* was smashed into small pieces and subjected to hydrodistillation with ultrapure water (3.5 L) in a Clevenger-type apparatus for approximately 4 h until translucent distillate was obtained. After flushing the condenser with ether, when no droplet of residue was attached to the arm of the Clevenger-type apparatus, the essential oil was collected and then dried over anhydrous sodium sulfate and Termovap Sample Concentrator. The obtained EO joined the former into glass flasks and was stored at a low temperature (−4 °C) for further analysis.

### 2.3. GC-MS and GC-FID Analysis

GC-MS analysis was carried out using an Agilent 7890–5975C gas chromatograph–mass spectrometer equipped with a fused silica capillary column type HP-5MS (30 m × 0.25 mm with film thickness, 0.25 microns, Agilent Technologies, Santa Clara, CA, USA). The interface temperature was 280 °C, and the injector temperature was 260 °C. The oven temperature was initially 50 °C. This was maintained for 4 min, programmed from 50 °C to 280 °C at a rate of 6 °C/min, and held steady for 3 min. Helium was used as the carrier gas at a 1.1 mL/min velocity. The mass spectrometer conditions were as follows: electron impact (EI) mode (electron energy = 70 eV), a scan range of 25–500 amu, a scan rate of 4.0 scans/s, and a quadrupole temperature of 150 °C. A 1% *w/v* sample solution in n-hexane was prepared, and 0.3 μL was injected using splitless mode [21,22]. GC-FID analysis was performed using an Agilent 7890 gas chromatograph with a type HP-5 fused silica capillary column (30 m × 0.25 mm with film thickness of 0.25 microns, Agilent Technologies, USA). The injector temperature was 260 °C, and the detector temperature was 305 °C. The oven temperature was initially 50 °C, maintained for 4 min, and then programmed from 50 °C to 280 °C at a rate of 6 °C/min and held steady for 3 min. Helium was used as the carrier gas at a 1.1 mL/min velocity. Identifying those compounds in EO is primarily based on comparing mass spectrometric data and Kovat retention indices relating to retention time with commercial libraries (NIST 20 and Adams) [23]. Specifically, the Kovat retention indices were calculated using a series of n-alkanes (C_8_–C_30_) with linear interpolation.

### 2.4. Antioxidant Activities Determination

#### 2.4.1. DPPH Method

The DPPH (2,2-diphenyl-1-picrylhydrazyl) ethanolic solution was prepared at a concentration of 0.17 mM. During DPPH free-radical scavenging activity evaluation, 96-well microplates were prepared. Following the addition of 200 µL of the stock DPPH solution, 50 µL of each dilution of the oil in ethanol (50, 25, 10, 5, 2.5, 1, and 0.5 mg mL^−1^) was added. For comparison, the positive solution was prepared with BHT (butylated hydroxytoluene) or Trolox but without the essential oil. The control was prepared with 200 µL of the stock DPPH solution and 50 µL of ethanol, and the sample blank was prepared with 50 µL of essential oil solution and 200µL of ethanol. After 30 min, readings were taken using a microplate reader (Epoch, Biotech company, Minneapolis, MN, USA) at a wavelength of 517 nm [24]. The DPPH free-radical scavenging capacity was calculated using the following equation:$$RSA\%=\left(1-\frac{A_{Sample}-A_{Sample\;Blank}}{A_{Control}}\right)\times 100\%$$
where *RSA*% assesses the “radical scavenging activity” of the DPPH radical, *A_Sample_* corresponds to the absorbance of the solution in the microplate with the sample at different concentrations, and *A_Sample Blank_* is the absorbance of ethanol without DPPH. *A_Control_* is the DPPH solution without the EO sample. After measuring the *RSA*% at each concentration, IC_50_ can be calculated.

#### 2.4.2. ABTS Method

The antioxidant of *L. nervosa* EO against an ABTS (2,2′-Azinobis-(3-ethylbenzothiazolin-6-sulfonic acid) diammonium salt) radical was investigated using the following method: the ABTS radical cation reagent was prepared by mixing 7 mM ABTS solution with 2.6 mM potassium persulfate, which was subsequently kept in the dark at room temperature for at least 12 h [25]. The ethanolic EO solution was prepared according to the following gradients: 25, 10, 5, 2.5, 1, and 0.5 mg mL^−1^. The absorbance of the solution was determined as 734 nm seven minutes into the reaction using a microplate reader (Epoch, Biotech company, USA). Ethanol was used as a blank and gradient-diluted solvent. BHT and Trolox served as positive controls. The scavenging capacity was calculated using the equation below:$$Inhibition\%=\frac{A_{0}-A}{A_{0}}\times 100\%$$
where *A*_0_ is the absorbance of 200 μL of diluted ABTS^+^ solution mixed with 50 μL of ethanol at 734 nm, while *A* is the absorbance of 200 μL diluted ABTS^+^ solution mixed with 50 μL of the sample solution at 734 nm. IC_50_ was then calculated [26].

#### 2.4.3. Ferric-Reducing Antioxidant Power (FRAP) Method

FRAP was assayed according to a previous procedure with some modifications [27]. The stock solutions included (1) pH 3.6 acetate buffer solution, (2) 10 mmol/L TPTZ (2,4,6-tripyridyl-s-triazine) solution, and (3) 20 mmol L^−1^ Fe^3+^ solution. The stock solutions were mixed at the proportion of 10: 1: 1 and were diluted 50 times with ethanol. Next, 50 μL of different dilutions of EOs (4000, 2000, 1000, 500, 250, and 100 μg mL^−1^) and 0.25 mg/mL Trolox solution (2, 5, 10, 15, and 20 μL) were mixed with 200 μL FRAP working reagent in a 96-well microplate. The blank solution was similarly prepared by replacing the EOs with ethanol. All tests were performed in triplicate and the results were averaged. After 30 min of reaction time, the absorbance of the resulting solution was measured at 593 nm using a microplate reader (Epoch, Biotech company, USA) [28].

### 2.5. Anti-Acetylcholinesterase Activity Test

The acetylcholinesterase inhibitory effect was measured using the previously described method with minor modifications [29]. Then, 140 μL of 0.1 mM, pH 8.0 phosphate-buffered saline (PBS), 20 μL of sample, and 20 μL of acetylcholinesterase solution containing 0.28 U/mL were mixed in a microplate and left to incubate at 4 °C for 20 min. Subsequently, 10 μL of 15 mM AChI (acetylthiocholine iodide) and 10 μL of 2 mM DTNB (5,5′-Dithiobis-(2-nitrobenzoic acid)) were added and then incubated for 20 min. The absorbance at 412 nm was read. A blank reaction was carried out using a PBS solution instead of the EO sample. Next, a complete inhibition reaction was performed by substituting the EO with huperzine A (100 μg/mL), and the sample blank reaction substituted AchE solution with PBS. The acetylcholinesterase inhibition rate is calculated using the following formula:*Inhibition*% = [(*B* − *C*) − (*S* − *SB*)]/(*B* − *C*)
where *B* is the absorbance of the blank reaction, *C* is the absorbance of the complete inhibition reaction, *S* is the EO-containing reaction’s absorbance, and *SB* is the absorbance of the sample blank reaction. The tests were carried out in triplicate, and we calculated the IC_50_ value then. Huperzine-A was used in the control as a positive reference.

## 3. Results and Discussion

### 3.1. EO Yield and Component Analysis

The essential oils of the plant were extracted using a Clevenger-type apparatus. A total of 0.3 mL essential oil was obtained from 750 g *L. nervosa* biomass via hydrodistillation (0.04 *v*/*w*). In a previous study of Rutaceae plant species, Aegle mamelons, the essential oil yield was 0.53%, which is much higher than *L. nervosa* [30]. However, the EO yield is largely determined by the phylogenetic position of the plant. As for Orchidaceae plants, research on four orchid species showed evaluated yields of 0.03%, 0.02%, 0.52%, and 0.10%, which are akin to the results of the current study [31]. Moreover, a study of *Cymbidium sinense* found a yield of 0.09% [32]. The yields of other Orchidaceae plants, specifically *Orchids sphegodes* and *Orchids purpurea*, were 0.05% and 0.02%, respectively [33]. Collectively, the relatively low EO yield of *L. nervosa* is reasonable according to previous studies of the same plant family. The essential oils of *L. nervosa* are yellow, hydrophobic, and have a unique scent. The total ion chromatogram of *L. nervosa* essential oil is shown in Figure 1.

The identities, relative percentages, retention indices (RI), CAS ID, and identification method of each component are presented in Table 1. Sixty-seven compounds representing 98.50% of the EO were identified from GC-MS and GC-FID analysis. Gas chromatographic analysis of the essential oil revealed methyl (9*E*,11*E*)-octadeca-9,11-dienoate (31.69%), n-hexadecanoic acid (15.08%), isopropyl palmitate (12.44%), propyl tetradecanoate (7.20%), tetradecanoic acid (4.01%), 17-octadecynoic acid (3.71%), and pentacosane (2.24%) to be the major constituents.

It is noteworthy that the EO composition is characterized by a primary amount of fatty acids and fatty acid esters (such as methyl (9*E*,11*E*)-octadeca-9,11-dienoate, *n*-hexadecanoic acid, and isopropyl palmitate). The abundance of those compounds indicated that the oil was derived from fatty acid and fatty acid ester chemotypes, which are thought to possess many biological activities, including larvicidal, ovicidal, and repellency activities, as well as a remarkable inhibitory effect in AChE, offering a research direction for further biological activity exploration [34,35]. However, the record of biological activities of (9*E*,11*E*)-octadeca-9,11-dienoate (the most abundant component in *L. nervosa* essential oil) is still lacking in the literature. The next most abundant compound was n-hexadecanoic acid, one of the most common fatty acids occurring in natural fats and oils and an active ingredient in the product ‘pneumotox’. This compound showed cytotoxicity to human leukemic cells but no cytotoxicity to normal human dermal fibroblast (HDF) cells [36]. Moreover, palmitic acid previously exhibited AChE inhibition and larval toxicity [37,38]. The results of other studies indicate that the third major component, isopropyl palmitate, is a model chemical penetration enhancer in drug release systems [39]. Taken together, these results indicate that essential oils from *L. nervosa* could be considered to possess a wide spectrum of biological activities. However, chemical components of plants’ EO are the result of the combined effects of many factors, including genetics, climate, edaphic, topography, elevation, and crosstalk, as well as the interactions among these factors [40,41].

### 3.2. Antioxidant Activities Evaluation

Different assays were introduced to measure the antioxidant capacity of our biological samples. According to a previous document clustering study, 2,2-diphenyl-1-picrylhydrazyl (DPPH)-based antioxidant capacity assays, together with ABTS-based and ferric-reducing antioxidant power (FRAP) are the most popular methods. All three are routinely practiced in research laboratories throughout the world. [42] This study evaluates the in vitro antioxidant activities of the EO from *L. nervosa* using DPPH, ABTS, and FRAP assays. The antioxidant values of the three methods are displayed in Table 2.

Recently, the spectrophotometric assays in the present study have been adopted to measure antioxidant capacity. The three assays in this study employ the same principle: a redox-active compound or synthetic-colored radical is generated, and the radical scavenging ability or redox-active compound reducing the activity of an EO is monitored using a spectrophotometer, with an appropriate standard applied to quantify antioxidant capacity [43].

The DPPH discoloration test is widely employed to evaluate natural products’ free-radical scavenging and antioxidant activities. The action between different free-radical scavengers with DPPH may be different. In plant essential oils, chemical compounds containing conjugated double bonds play a major role in scavenging free radicals [44]. Terpenoids can rapidly terminate DPPH free-radical chain reactions since they contain conjugated double bonds with strong chain-breaking antioxidant activity. However, the proportion of these compounds in *L. nervosa* essential oil is relatively small, so the result showed a relatively low DPPH scavenging activity with 44.2 ± 2.9% scavenging ability at 10 mg/mL concentration. Its potency seems weaker than those obtained from other species of essential oil [45,46,47].

It was also observed that the EO samples exhibit relatively high ABTS values (IC_50_ 721.95 ± 9.93 μg/mL). *L. nervosa* essential oil increased in a sigmoidal dose-dependent manner over the concentration range tested in the ABTS assay, as shown in Figure 2. The present results of the ABTS test were better than those obtained from the DPPH test, in accordance with previous research based on the analysis of a large number of food samples. This demonstrates that the ABTS assay can more accurately estimate antioxidant capacity compared to the DPPH assay [43].

An FRAP method using reductants via a redox-linked colorimetric method that employs an easily reduced oxidant in stoichiometric excess could be a simple way of assessing the total antioxidant ability [48]. This assay is simple and quick to perform, and the reaction has a linear relationship with the molar concentration of the present antioxidants [48]. The reducing capacity is related to the degree of hydroxylation and the degree of conjugation of the bonds present in the phenolic compounds found in EOs [49]. The essential oil demonstrated moderate ferric-ion-reducing activity (Trolox equivalent antioxidant concentration).

Generally, the antioxidant activities of essential oils are highly dependent on their chemical composition and content. On the other hand, essential oils are complex mixtures and consist of many compounds; therefore, this complexity generally makes it difficult to characterize the activity pattern. Therefore, Prior et al. suggest using multiple assays to evaluate the overall antioxidant capacity and create an “antioxidant profile” that covers reactivity towards aqueous and lipid/organic radicals through various mechanisms. Using different assays with different mechanisms is recommended for complex samples to fully understand antioxidant action [50].

### 3.3. Acetylcholinesterase Inhibitory Effects

AChE (EC 3.1.1.7) consists of a complex protein of the α/β hydrolase fold type, generally with an ellipsoid shape containing a gorge about 20 Å deep [51]. As shown in Figure 3, the essential oil exhibited highly potent inhibition of the AChE enzyme. The resulting IC_50_ value was 51.96 ± 14.26 μg/mL, demonstrating effective inhibitory activity against AChE, which is significantly higher than that of essential oils distilled from other species (IC_50_ = 220, 175, 570, and 152 μg/mL, four *Ocimum* species) and the reversible inhibitor, physostigmine (IC_50_ 270 μg/mL) [52]. A previous report found that the EO in the leaves and roots of *Cymbopogon schoenanthus* had an IC_50_ value of 260 μg/mL, which is also weaker than inhibitory *L. nervosa* EO [53]. However, a study on EO in *Nepeta menthoides* showed desirable anti-acetylcholinesterase activity (IC_50_ of 64.87 μg/mL), similar to the data obtained in this study [54].

Due to the desired enzyme inhibitory effect of *L. nervosa* essential oil, it was necessary to determine the active component responsible for the anti-acetylcholinesterase activity. The active site of this enzyme is composed of peripheral tryptophan and phenylalanine, tyrosine residues located at the entrance of the gorge. Enzyme amino acid residues may interact with compounds containing multiple aromatic rings due to the affinity of those molecules to residues localized at the entrance of the active site [55].

However, further molecular docking is necessary to reveal how compounds bind to key amino acids in the catalytic domain of AChE.

## 4. Conclusions

In conclusion, the present study found that the essential oil obtained from *L. nervosa* has significant anti-acetylcholinesterase activity and possesses moderate antioxidant activity, as evaluated using ABTS and FRAP assays. The results show that this essential oil could be considered a natural source for isolating active constituents for food preservatives and therapeutic applications of AD. However, these results cannot be used to determine which single component is responsible for most biological activity. Therefore, more research on structural chemistry and computational chemistry, as well as further in vivo investigations, is required. 

## Figures and Tables

**Figure 1 biomolecules-13-01089-f001:**
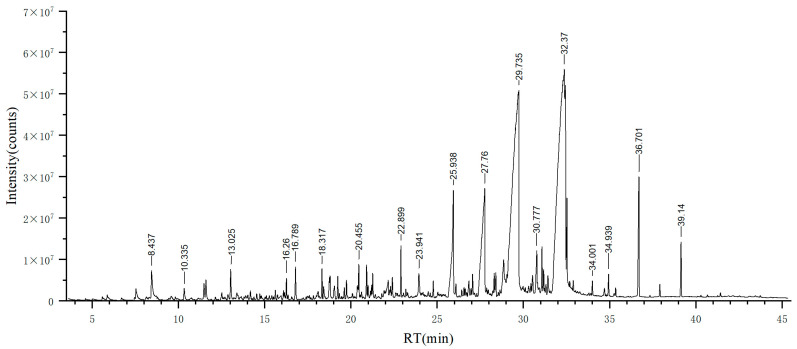
Total ion chromatogram of EO from *L. nervosa* derived from GC–MS.

**Figure 2 biomolecules-13-01089-f002:**
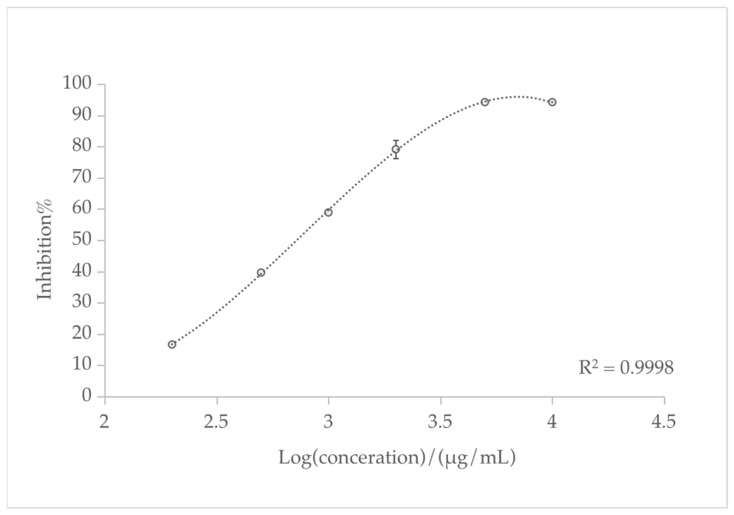
Concentration-dependent ABTS scavenging activity of *L. nervosa* essential oil.

**Figure 3 biomolecules-13-01089-f003:**
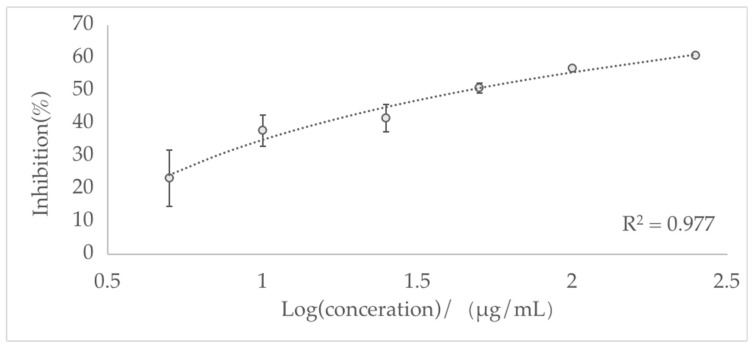
The concentration-dependent anti-acetylcholinesterase activity of *L. nervosa* essential oil.

**Table 1 biomolecules-13-01089-t001:** Chemical composition of EOs distilled from *L. nervosa*.

No.	Retention Time, tR (min)	Compound	RI ^a^	RI ^b^	Area (%)	Identification Method	CAS ID
1	5.867	Heptanal	907	901	0.22%	RRI, MS	111-71-7
2	7.531	Benzaldehyde	968	962	0.43%	RRI, MS	100-52-7
3	8.437	2-Pentylfuran	995	993	1.08%	RRI, MS	3777-69-3
4	9.583	2-Octyn-1-ol	1038	-	0.14%	MS	20739-58-6
5	10.335	(*E*)-2-Octenal	1065	1060	0.25%	RRI, MS	2548-87-0
6	11.481	Linalool	1103	1099	0.29%	RRI, MS	78-70-6
7	11.59	Nonanal	1108	1104	0.39%	RRI, MS	124-19-6
8	12.507	3-Nonen-2-one	1127	1142	0.17%	RRI, MS	14309-57-0
9	12.872	Cucumber aldehyde	1159	1155	0.16%	RRI, MS	557-48-2
10	13.025	(*E*)-2-Nonenal	1165	1162	0.56%	RRI, MS	18829-56-6
11	13.39	4-Ethylbenzaldehyde	1178	1180	0.18%	RRI, MS	4748-78-1
12	14.171	Decanal	1209	1206	0.17%	RRI, MS	112-31-2
13	15.949	Nonanoic acid	1283	1273	0.17%	RRI, MS	112-05-0
14	16.091	Anethole	1290	1286	0.14%	RRI, MS	104-46-1
15	16.26	2-Undecanone	1296	1294	0.31%	RRI, MS	112-12-9
16	16.789	(*E*,*E*)-2,4-Decadienal	1320	1317	0.48%	RRI, MS	25152-84-5
17	18.093	*n*-Decanoic acid	1379	1373	0.19%	RRI, MS	334-48-5
18	18.317	β-Damascenone	1388	1386	0.39%	RRI, MS	23726-93-4
19	18.419	(+)-Sativen	1393	1396	0.23%	RRI, MS	3650-28-0
20	18.748	β-Longipinene	1408	1403	0.33%	RRI, MS	41432-70-6
21	18.791	Longifolene	1411	1405	0.34%	RRI, MS	475-20-7
22	19.048	Dihydrodehydro-β- ionone	1424	1424	0.32%	RRI, MS	20483-36-7
23	19.239	α-Ionone	1433	1426	0.31%	RRI, MS	127-41-3
24	19.615	Acenaphthylene	1451	1454	0.15%	RRI, MS	208-96-8
25	19.74	Dihydropseudoionone	1457	1456	0.25%	RRI, MS	689-67-8
26	20.373	Curcumene	1487	1483	0.27%	RRI, MS	644-30-4
27	20.455	β-Ionone	1491	1491	0.45%	RRI, MS	14901-07-6
28	20.908	Tridecanal	1513	1512	0.47%	RRI, MS	10486-19-8
29	20.990	Dibenzofuran	1518	1514	0.15%	RRI, MS	132-64-9
30	21.197	δ-Cadinene	1528	1524	0.13%	RRI, MS	483-76-1
31	22.163	Dodecanoic acid	1577	1568	0.55%	RRI, MS	143-07-7
32	22.304	Fluorene	1584	1583	0.15%	RRI, MS	86-73-7
33	22.403	(*Z*)-α-Bisabolene epoxide	1589	1586	0.30%	RRI, MS	111536-37-9
34	22.899	Tetradecanal	1615	1613	0.66%	RRI, MS	124-25-4
35	23.183	Oxacyclotetradeca-4,11-diyne	1630	1639	0.14%	RRI, MS	6568-32-7
36	23.941	Tridecanoic acid	1671	1666	0.63%	RRI, MS	638-53-9
37	24.776	Pentadecanal	1717	1715	0.23%	RRI, MS	2765-11-9
38	25.938	Tetradecanoic acid	1782	1768	4.01%	RRI, MS	544-63-8
39	26.085	Phenanthrene	1790	1775	0.22%	RRI, MS	85-01-8
40	26.56	Hexadecanal	1818	1817	0.13%	RRI, MS	629-80-1
41	26.843	Isopropyl myristate	1835	1827	0.32%	MS	117-27-0
42	27.051	Hexahydrofarnesyl acetone	1847	1844	0.35%	RRI, MS	502-69-2
43	27.76	Propyl tetradecanoate	1889	1896	7.20%	RRI, MS	14303-70-9
44	28.311	Farnesyl acetone	1923	1919	0.33%	RRI, MS	1117-52-8
45	28.404	Methyl palmitate	1929	1926	0.34%	RRI, MS	112-39-0
46	28.856	11-Hexadecenoic acid	1957	1953	1.66%	RRI, MS	2416-20-8
47	29.042	Dodecenyl succinic anhydride	1969	1968	0.36%	RRI, MS	19780-11-1
48	29.538	*n*-Hexadecanoic acid	-	1970	15.08%	MS	57-10-3
49	29.735	Isopropyl palmitate	2012	2023	12.44%	RRI, MS	142-91-6
50	30.531	Fluoranthene	2063	2054	0.37%	RRI, MS	206-44-0
51	30.777	Heptadecanoic acid	2079	2071	0.87%	RRI, MS	506-12-7
52	31.077	Methyl linoleate	2099	2092	0.67%	RRI, MS	112-63-0
53	31.175	Methyl linolenate	2105	2098	0.33%	RRI, MS	301-00-8
54	31.427	Phytol	2112	2114	0.39%	RRI, MS	150-86-7
55	32.370	methyl (9*E*,11*E*)-octadeca-9,11-dienoate	2186	2187	31.69%	RRI, MS	13038-47-6
56	32.441	17-Octadecynoic acid	2190	2199	3.71%	RRI, MS	34450-18-5
57	32.528	2-Methyl-*Z*,*Z*-3,13- octadecadienol	2196	-	1.62%	MS	519002-96-1
58	32.686	Isopropyl linolenate	2207	2200	0.20%	RRI, MS	83918-59-6
59	32.752	10-trans,12-cis-Linoleic acid	2213	2222	0.17%	RRI, MS	2420-56-6
60	32.893	2,4,5,7-Tetramethylphenanthrene	2223	-	0.21%	MS	7396-38-5
61	34.001	Tricosane	2300	2300	0.18%	RRI, MS	638-67-5
62	34.699	Diroleuton	2352	2346	0.19%	RRI, MS	1783-84-2
63	34.939	Octadecanamide	2370	2374	0.34%	RRI, MS	124-26-5
64	35.348	Tetracosane	2399	2400	0.13%	RRI, MS	646-31-1
65	36.701	Pentacosane	2503	2500	2.24%	RRI, MS	629-99-2
66	37.907	Hexacosane	2599	2600	0.18%	RRI, MS	630-01-3
67	39.140	Heptacosane	2699	2700	0.81%	RRI, MS	593-49-7

Concentration calculated from the total ion chromatogram. RI ^a^: Calculated retention index. RI ^b^: Retention index obtained from the mass spectral database. RRI: Relative retention indices calculated against *n*-alkanes. The identification method is based on the relative retention indices (RRI) of authentic compounds on the HP-5MS column. MS was identified based on computer matching of the mass spectra with the NIST/EPA/NIH 2020 Mass Spectral Database, Essential Oils GC/MS Library (Version 4, Robert Adams), and comparison with the literature data.

**Table 2 biomolecules-13-01089-t002:** Antioxidant activities of *L. nervosa* essential oil expressed as IC_50_ values for DPPH, ABTS, and the antioxidant ability of FRAP assays.

Tested Samples	DPPH 50% Effective Concentration (μg/mL)	ABTS 50% Effective Concentration (μg/mL)	FRAP Antioxidant Capacity (μM/g)
*L. nervosa* EOs	>10,000	748.3	39.64 ± 3.38
BHT	37.02 ± 2.21	14.69 ± 1.32	-
Trolox	18.23 ± 1.12	9.47 ± 1.21	-

## Data Availability

The data presented in this study are available from the corresponding author upon request.

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
