# Peer review of "Chemical Composition, In Vitro Antioxidant Activities, and Inhibitory Effects of the Acetylcholinesterase of Liparis nervosa (Thunb.) Lindl. Essential Oil"

_biomolecules, 2023, doi:10.3390/biom13071089_

Round 1

Reviewer 1 Report

This manuscript is on the essential oil composition of Liparis nervosa (Thunb.) Lindl. grown in China and determining their antioxidant and inhibitory effect for Acetylcholinesterase. It is an interesting subject which is investigated from different sides. However, it needs some revision before acceptance. Abstract section is fine, but in Introduction section below comments are suggested:

Please write the scientific description for clearing heat fire and cooling blood.

One paragraph should not be one sentence. Rewrite and clarify this paragraph: Much scientific evidence on the therapeutic effects of plant-derived molecules can be adduced, and exploring the natural product repository has led to the development of some conventional blockbuster medicines, such as the anticancer drug Taxol [3].

in vivo should be in Italic.

In the materials and methods section, it would be so good to use and internal standard in GC analysis to quantify the essential oil compounds rather than presentation by the percentages.

Please compare the results with previously published data, as the essential oil amount obtained in this study is very low (0.3ml per 750g). Discuss it what have caused this low yield.

Peak eluted in 29.707 and 32.342 min was not mentioned in the Table 1, as they are in high ratio and presented in chromatogram.

Some writing issues are suggested in the comments to the authors section.

Author Response

Dear Editor and Reviewer 1,

Thank you for the review of our manuscript "Chemical Composition, in vitro Antioxidant Activities and In-hibitory Effect for Acetylcholinesterase of Liparis nervosa (Thunb.) Lindl. Essential Oil". Your constructive comments are appreciated very much. We have responded to the comments point by point and made revisions accordingly.

Reviewer 1:

This manuscript is on the essential oil composition of Liparis nervosa (Thunb.) Lindl. grown in China and determining their antioxidant and inhibitory effect for Acetylcholinesterase. It is an interesting subject which is investigated from different sides. However, it needs some revision before acceptance. Abstract section is fine, but in Introduction section below comments are suggested:

(1) Please write the scientific description for clearing heat fire and cooling blood.

Response: Thanks for your comment. The meaning of clearing heat fire and cooling blood refers to the essential oil can eliminate excessive heat, alleviate inflammation, and regulate blood circulation,such as the velocity of blood in the field of Chinese medicine.

(2) One paragraph should not be one sentence. Rewrite and clarify this paragraph: Much scientific evidence on the therapeutic effects of plant-derived molecules can be adduced, and exploring the natural product repository has led to the development of some conventional blockbuster medicines, such as the anticancer drug Taxol [3].

Response: Thanks for your reminder. There were some problems with the submission, which resulted in this sentence being a separate paragraph. And we have revised ths sentence to form a paragraph together with the following sentences.

(3) in vivo should be in Italic.

Response: Thanks for your careful inspection. We have modified this mistake in the article.

(4) In the materials and methods section, it would be so good to use and internal standard in GC analysis to quantify the essential oil compounds rather than presentation by the percentages.

Response: The content of each component measured by GC analysis in this paper is relative content, and the normalization of peak area was used. Therefore, no internal standard was used.

(5) Please compare the results with previously published data, as the essential oil amount obtained in this study is very low (0.3ml per 750g). Discuss it what have caused this low yield.

Response:  For the low yield of esstion oil, the reasons are added in the discussion part, to be specifically, there are intrinsic and external factors affecting the EO yield.

[1] Intrinsic factors:

Plant phylogenetic position: according to our experiences, Asteraceae, Brassicaceae, Lauraceae, Lamiaceae plants often possess rich essential oils. For example, In previous research on Hyptis villosa (Lamiaceae), the oil yield was 0.2% (https://doi.org/10.1080/10412905.2013.828327). However, based on our previous studies, some plant families indeed have relatively low EO yield such as Poaceae, Fabaceae, Orchidaceae …… Our plant material is L. nervosa, which is a species in Orchidaceae family. In a previous study on four orchid species, the EO yields were evaluated as 0.03%, 0.02%, 0.52% and 0.10% (https://doi.org/10.3390/molecules24213878). In this study, the EO yield was about 0.04%. Hence EO yield reasonable.

[2] External factors:

Drying method: The plant materials were dried in the sun, so it would cause some loss of the volatile compounds. That’s a reason why the yield is low.

(6) Peak eluted in 29.707 and 32.342 min was not mentioned in the Table 1, as they are in high ratio and presented in chromatogram.

Response: Thanks for your comments. We revised the Figure 1 in the paper, because when we redawing the total ion-chromatogram of L. nervosa derived from GC–MS, the annotations are not locked to the maximum. So it is a labeling error.

Sincerely,

Jiayi Zhao

On behalf of all co-authors of the paper

Reviewer 2 Report

- The sentences in lines 12-14 and 37-39 should be corrected

- In line 82 should be "hydrodistillation"

- In line 89 should be " gas chromatograph"

- In line 92 should be " injector temperature"

- In line 96 should be "25-500 amu"

- In line 97 should be "quadrupole"

- In line 98 instead of " no splitting" should be "splitless"

- The GC-FID conditions should be described

- The retention indices calculation method should be described

- In line 172 instead of "index" should be "indices"

- In line 173 instead of "67" should be "sixty seven"

- In Table 1 instead of "RT" should be "Retention time, tR (min)"

- In Table 1 instead of  "Nist" should be "NIST"

- In Table 1 should be:

2-Pentylfuran

4-Ethylbenzaldehyde

2,4, 5,7-Tetramethylphenanthrene

Some sentences should be corrected.

Author Response

Dear Editor and Reviewer 2,

Thank you all for the review of our manuscript “Chemical Composition, in vitro Antioxidant Activities and In-hibitory Effect for Acetylcholinesterase of Liparis nervosa (Thunb.) Lindl. Essential Oil”. Your constructive comments are appreciated very much. We have responded to the comments and made modifications accordingly.

Reviewer 2:

- The sentences in lines 12-14 and 37-39 should be corrected

Response: Thanks a lot for your comment. Sentences in line 12-14 were corrected: “The essential oil was obtained by hydrodistillation, and compounds chemicals of that was ana-lyzed by GC-MS and GC-FID. 2, 20-azino-bis (3-ethylbenzothiazoline-6-sulfonic acid) (ABTS), 2,2-diphenyl-1-picrylhydrazyl (DPPH), and ferric reducing assay power (FRAP) methods were used to evaluate antioxidant activity.” Sentences in lines 37-39 were corrected as “Therefore, traditional herbal medicine is gaining growing popularity and being widely embraced in treatment approaches.”

- In line 82 should be "hydrodistillation"

- In line 89 should be " gas chromatograph"

- In line 92 should be " injector temperature"

- In line 96 should be "25-500 amu"

- In line 97 should be "quadrupole"

- In line 98 instead of " no splitting" should be "splitless"

Response: Thanks a lot for the aforesaid comments and modification concerning the spelling and mistakes were corrected in paper.

- The GC-FID conditions should be described

Response: Thanks a lot for the comments. GC-FID analysis was performed on an Agilent 7890 gas chromatograph with a fused silica capillary column type HP-5 (30 m × 0.25 mm with film thickness 0.25 microns, Agilent Technologies, USA). The injector temperature was 260 °C, and the detector temperature was 305 °C. The oven temperature was initially 50 °C and held stable for 4 min, and then programmed from 50°C to 280 °C at the rate of 6°C/min and held steady for 3 min. Helium was used as the carrier gas at a 1.1 mL/min velocity.

- The retention indices calculation method should be described

Response: Thanks for your comments. Retention indices (RI) is used to calculate the relative retention time of series of n-alkanes (C8-C30) with linear interpolation by Kovat.

The formula of RI is:

- In line 172 instead of "index" should be "indices"

- In line 173 instead of "67" should be "sixty seven"

- In Table 1 instead of "RT" should be "Retention time, tR (min)"

- In Table 1 instead of  "Nist" should be "NIST"

- In Table 1 should be:

2-Pentylfuran

4-Ethylbenzaldehyde

2,4, 5,7-Tetramethylphenanthrene

Response: Thanks a lot for the comments. We have modified all the the spelling mentioned.

Sincerely,

Jiayi Zhao

On behalf of all co-authors of the paper
